

# Population-level call properties of endangered *Dryophytes suweonensis sensu lato* (Anura: Amphibia) in South Korea

Md Mizanur Rahman[1], Jiyoung Yun[2], KaHyun Lee[2], Seung-Ha Lee[1], Seung-Min Park[2], Choong-Ho Ham[2] and Ha-Cheol Sung[1,3]

[1] Department of Biological Sciences, Chonnam National University, Gwangju, South Korea
[2] Department of Biological Sciences and Biotechnology, Chonnam National University, Gwangju, South Korea
[3] Research Center of Ecomimetics, Chonnam National University, Gwangju, Republic of Korea

Corresponding author
Ha-Cheol Sung,
shcol2002@chonnam.ac.kr

## ABSTRACT

Calling is one of the unique amphibian characteristics that facilitates social communication and shows individuality; however, it also makes them vulnerable to predators. Researchers use amphibian call properties to study their population status, ecology, and behavior. This research scope has recently broadened to species identification and taxonomy. *Dryophytes flaviventris* has been separated from the endangered anuran species, *D. suweonensis*, based on small variations in genetic, morphometric, and temporal call properties observed in South Korea. The Chilgap Mountain (CM) was considered as the potential geographic barrier for the speciation. However, it initiated taxonomic debates as CM has been hardly used and is considered a potential barrier for other species. The calls of populations from both sides are also apparently similar. Thus, to verify the differences in call properties among populations of *D. suweonensis sensu* lato (s.l.; both of the species), we sampled and analyzed call data from five localities covering its distribution range, including the southern (S) and northern (N) parts of CM. We found significant differences in many call properties among populations; however, no specific pattern was observed. Some geographically close populations, such as Iksan (S), Wanju (S), and Gunsan (S), had significant differences, whereas many distant populations, such as Pyeongtaek (N) and Wanju (S), had no significant differences. Considering the goal of this study was only to observe the call properties, we cautiously conclude that the differences are at the population level rather than the species level. Our study indicates the necessity of further investigation into the specific status of *D. flaviventris* using robust integrated taxonomic approaches, including genetic and morphological parameters from a broader array of localities.

## INTRODUCTION

Amphibians are one of the world's most vulnerable vertebrate animal groups (*IUCN, 2023*). Although the amphibian diversity of many regions is underestimated and suffering from a lack of information (*Rahman et al., 2019*; *Rahman et al., 2022*), the introduction of integrated traditional and modern molecular taxonomic approaches boosted the trends of

deiscovering new records and new species descriptions (*Rahman et al., 2020a*; *Rahman et al., 2020b*; *Vences et al., 2023*). Despite this progress creating a few taxonomic debates and ambiguity in classification (*Sanchez et al., 2018*), the number of total amphibian species in the world has increased manifolds (*AmphibiaWeb, 2023*). The trend of new species description sometimes misleads the policymakers on amphibian conservation issues. It is already evident that amphibians across the world are facing immense conservation challenges from diverse factors, including climate change, temperature fluctuation (*Grimm et al., 2008*; *Dawson et al., 2011*; *Li, Cohen & Rohr, 2013*; *Rahman, 2014*; *Zhao et al., 2022*), emerging infectious diseases (*Grant et al., 2016*; *Cohen et al., 2018*; *Scheele et al., 2019*), invasive species (*Nunes et al., 2019*; *Falaschi et al., 2020*; *Park et al., 2022*), and habitat alteration (*Cushman, 2006*; *Becker et al., 2007*; *Decena et al., 2020*). Additionally, the unique characteristics of amphibians make them susceptible to declining risks from both aquatic and terrestrial ecosystems while also enabling them to thrive in both environments (*Becker & Loyola, 2007*). For instance, amphibian characteristics such as the biphasic life cycle, environmental sensitivity, ectothermic nature, permeable skin, *etc.*, make this animal group vulnerable to many fatal diseases (*Campbell et al., 2012*; *Burraco et al., 2020*). On the other hand, the unique characteristics of amphibians, such as their calling behavior, assist coordinating their seasonal and daily activities and facilitate individual and social communications (*Kelley, 2004*; *Natale et al., 2010*).  Frog calling provides basic information on their physical and environmental status (*Snowdon, 2011*). They also use it to defend their territory and attract partners for breeding (*Duellman, 1970*; *Wells, 2007*; *Wells & Schwartz, 2007*). Given the necessity of attracting conspecific individuals, researchers suggested the possibility of using the frog calls in their taxonomy (*Ryan & Rand, 2001*; *Wells & Schwartz, 2007*). Thus, in addition to using traditional and molecular techniques (*Rahman et al., 2020a*; *Rahman et al., 2020b*; *Rahman et al., 2022*; *Nneji et al., 2021*), researchers are increasingly employing call properties in amphibian species identification and taxonomy (*Kohler et al., 2017*; *Borzée et al., 2020*).

Although the use of call properties in amphibian taxonomy is becoming common, there is evidence of considerable variations in call properties among individuals, populations, and environmental conditions (*Kaefer & Lima, 2012*; *Kaefer, Tsuji-Nishikido & Lima, 2012*; *Velasquez, 2014*; *Forti, Marquez & Bertoluci, 2015*; *Forti, Lingnau & Bertoluci, 2017*). Some studies also showed that the calls could change after training during the juvenile stages (*Dawson & Ryan, 2009*). Health conditions may also influence call properties (*Kelley, 2004*). Particularly, the temporal call properties can change considerably depending on environmental conditions and the physical status of the individuals (*Wong et al., 2004*; *Lingnau & Bastos, 2007*). Hence, we need further studies on the comparison of frog call properties to enhance the accuracy of their taxonomic uses (*Forti, Martins & Bertoluci, 2012*; *Forti, Marquez & Bertoluci, 2015*; *Forti, Lingnau & Bertoluci, 2017*; *Hepp, Lourenco & Pombal Jr, 2017*). Nevertheless, call properties can be helpful in amphibian taxonomy and species identification when applied with proper guidelines and a comprehensive understanding of congeneric species (*Kohler et al., 2017*).

Recently, *Borzée et al. (2020)* separated *Dryophytes flaviventris* from *D. suweonensis* based on narrow genetic and morphometric differences along with variations in temporal call
properties observed in South Korea, initiating a taxonomic debate. Although there is hardly any prior report on the significant role of Chilgap Mountain (CM) ranges on speciation, *Borzée et al. (2020)* suggested that it could have acted as the geographic barrier for these two species. However, the calls of populations from both sides are apparently similar. Thus, to explore and verify the differences in call properties among populations of *D. suweonensis* sensu lato (s.l.; both of the species), we analyzed call data from five localities covering its distribution range (based on *Borzée et al., 2020*), including the southern (S) and northern (N) parts of CM. The results indicated variations at the population-level rather than the species-level. Hence, our study suggests the need for further justification of the specific status of *D. flaviventris* using robust taxonomic approaches and including more samples from more localities.

## METHODS

The *D. suweonensis* s.l. call data were collected from various locations in South Korea, including Eumseong (May–August 2012), Pyeongteak (May 2021, and June 2022), Iksan (May 2021, and June 2022), Wanju (June 2022), and Gunsan (May 2021, and June 2022) (Fig. 1). Eumseong (36.9397°N, 127.6905°E) and Pyeongtaek (36.9921°N, 127.1129°E) are located to the north of CM (hereafter northern population; N), whereas Iksan (35.9483°N, 126.9576°E), Wanju (35.8913°N, 127.2539°E), and Gunsan (35.9677°N, 126.7366°E) are located to the south of CM (hereafter southern population; S). We did not require permits as we only recorded the calls and did not catch the individuals. The temperature during the data collection period varied from 16.6 °C to 26.1 °C and had no significant influence on the calls of this species. All calls were recorded from the evening to midnight. We collected call data from a total of 56 individuals, 25 from Eumseong, five from Pyeongteak, 12 from Iksan, seven from Wanju, and seven from Gunsan. Calls were recorded using recorder PMD661 MKIII mic: Sennheiser MKE600. Following the suggestions from *Kohler et al. (2017)*, both temporal and spectral domains were measured using Raven Pro 1.6.4 (Cornell Lab of Ornithology, New York, USA). We categorized the call properties and performed the subsequent analysis following the methods outlined by *Park, Jeong & Jang (2013)* and *Borzée et al. (2020)*. An advertisement call consists of a train of notes, and each note consists of a series of pulses (*McLister, Stevens & Bogart, 1995*). Thus, a single note is made up of a few independent pulses (single separate pulses) and a connected pulse (Fig. 2). Following these criteria, we measured call properties, like delta time (the difference between the start time and end time of the note; measured in seconds), number of independent pulses (single separate pulses ahead of the connected pulses in a note; counted in numbers), duration of connected pulses (the difference between the start time and end time of the connected pulses in a note; measured in seconds), internote interval (the difference between the end of a note and start of the next note; measured in seconds), low frequency (the lower frequency limit of the note; measured in Hz), high frequency (the upper frequency limit of the note; measured in Hz), max frequency (the frequency at which maximum power occurs in the

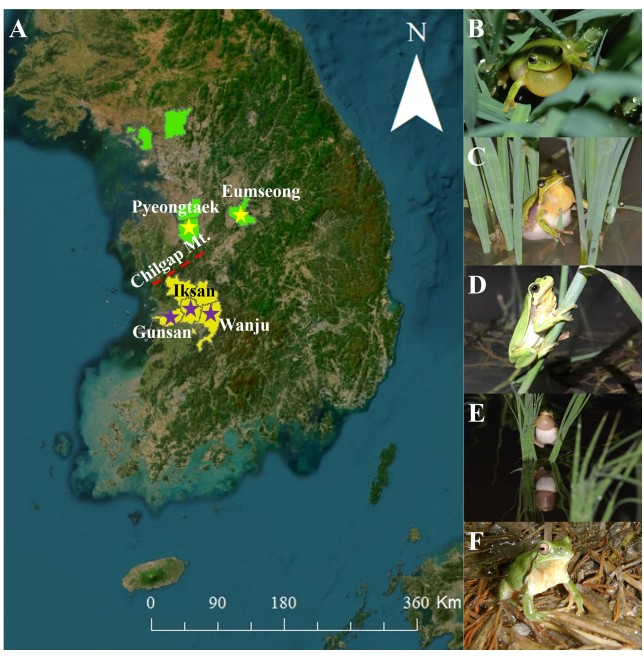

**Figure 1** **Study sites and species.** (A) Map of South Korea with the study sites. The 'yellow stars' indicate the collection localities of *Dryophytes suweonensis* and the 'purple stars' indicate the collection localities of *D. flaviventris*. The map was created using ArcMap (ver. 10.7, ESRI; https://support.esri.com/en/products/desktop/arcgis-desktop/arcmap/10-7-1) and QGIS Desktop TMS for Korean users Plugin (ver. 1.5; https://plugins.qgis.org/plugins/tmsforkorea/). (B) An individual from Eumseong. (C) An individual from Pyeongtaek. (D) An individual from Iksan. (E) An individual from Wanju. (F) An individual from Gunsan. The 'light green background' indicate the known distribution range of *D. suweonensis* and the 'yellow background' indicates the known distribution range of *D. flaviventris*.

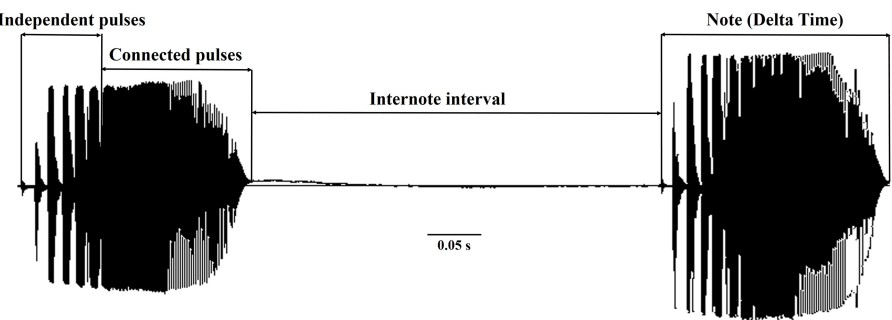

**Figure 2** **The temporal call properties of *Dryophytes suweonensis* sensu lato.**

note; measured in Hz), and 90% Bandwidth Frequency (the difference between the 5% and 95% frequencies in the note; measured in Hz).

After determining the call properties of each population, we used principal component analysis (PCA) to test variations in call properties between populations. We analyzed call properties from a total of 2,858 notes (on an average 51.04 notes for each individual), 468 from Iksan ($39.08 \pm 2.148$), 485 from Wanju ($69.29 \pm 28.45$), 914 from Gunsan

**Table 1 Variable loadings on principal components (PCs) and the subsequent results of ANOVA.** The bold fonts indicate loadings < 0.55 in the PCs and significant values in the results of ANOVA.

| Variables | PC1 | | PC2 | |
|---|---|---|---|---|
| Low Freq (Hz) | 0.29 | | **0.65** | |
| Delta time (s) | **0.92** | | 0.11 | |
| Max Freq (Hz) | −0.23 | | **0.83** | |
| Duration of connected pulse | **0.87** | | 0.18 | |
| Internote interval | **0.67** | | −0.24 | |
| Eigenvalues | 2.20 | | 1.20 | |
| % of variance | 43.98 | | 24.06 | |
| **ANOVA** | **Between group** | **Within group** | **Between group** | **Within group** |
| Sum of squares | 554.218 | 2,302.782 | 643.987 | 2,213.013 |
| *df* | 4 | 2,853 | 4 | 2,853 |
| Mean square | 138.554 | 0.807 | 160.997 | 0.776 |
| *F* | **171.660** | – | **207.556** | – |
| *p* | 0.000 | – | 0.000 | – |

(130.57 ± 67.183), 291 from Pyeongtaek (58.2 ± 16.684), and 700 notes from Eumseong (28 ± 1.502). The PCA was set to extract components if their Eigenvalue was > 1 under a varimax rotation. Variables were selected as loading into a PC if > 0.55. PC1 represented the temporal call properties, whereas PC2 represented the call frequencies (Table 1). Once the PCs were extracted, we performed ANOVA to identify significant differences within and between populations. After determining the significant differences between populations, we performed the *Post hoc* Tukey test based on Honesty Significance Difference (HSD) to reveal the pairwise variations between populations. The significance level was set at 0.05. All statistical tests were conducted using SPSS (SPSS, Inc., Chicago, USA).

# RESULTS

## Population-level call properties

The average low frequency of the calls varied from 1,001.92 (±7.79; Wanju) to 1,263.76 (±4.28; Gunsan) among the five populations. The average high frequencies of calls from the Iksan (S), Wanju (S), Gunsan (S), Pyeongtaek (N), and Eumseong (N) populations were 16,415.78 (±131.92), 16,134.16058 (±143.43), 8,780.40 (±119.03), 10,481.14 (±171.48), and 22,890.97 (±67.14), respectively. We measured the highest average max frequency from the Iksan (S) population (3,359.92 ± 9.78) and the lowest from the Eumseong (N) population (3,079.69 ± 27.59). Whereas, we got the maximum average of 90% Bandwidth of calls from the Iksan (S) population (2,748.15 ± 53.85) and the minimum from the Gunsan (S) population (1,804.84 ± 18.27; Table 2). Furthermore, the Gunsan (S) population had the highest average delta time (0.16 ±0.0004) and duration of connected pulses (0.10 ± 0.0003) among the temporal call properties. Eumseong (N) population was measured with the highest average internote interval (0.36 ± 0.005) and Iksan (S) population with the highest number of independent pulses (6.30 ± 0.05) (Table 2).

**Table 2  Call properties of Dryophytes suweonensis sensu lato from five localities.** Here, values after '±' indicate the standard error, 'S' indicate the populations located to the south of Chilgap Mountain, and 'N' indicate the populations located to the north of Chilgap Mountain.

| Call properties | Iksan (S) | Wanju (S) | Gunsan (S) | Pyeongtaek (N) | Eumseong (N) |
|---|---|---|---|---|---|
| Low Freq (Hz) | 1,227.44 ± 5.43 | 1,001.92 ± 7.79 | 1,263.76 ± 4.28 | 1,068.59 ± 7.38 | 1,109.73 ± 2.74 |
| High Freq (Hz) | 16,415.78 ± 131.92 | 16,134.16 ± 143.43 | 8,780.40 ± 119.03 | 10,481.14 ± 171.48 | 22,890.97 ± 67.14 |
| Max Freq (Hz) | 3,359.92 ± 9.78 | 3,251.38 ± 5.35 | 3,329.86 ± 4.61 | 3,194.63 ± 27.34 | 3,079.69 ± 27.59 |
| BW 90% (Hz) | 2,748.15 ± 53.85 | 2,119.93 ± 10.44 | 1,804.84 ± 18.27 | 1,903.60 ± 33.28 | 2,157.59 ± 15.54 |
| Delta Time (s) | 0.14 ± 0.0009 | 0.14 ± 0.0006 | 0.16 ± 0.0004 | 0.14 ± 0.001 | 0.15 ± 0.001 |
| Duration of connected pulses | 0.08 ± 0.0007 | 0.09 ± 0.0004 | 0.10 ± 0.0003 | 0.09 ± 0.0008 | 0.096 ± 0.0008 |
| Internote interval | 0.30 ± 0.004 | 0.26 ± 0.002 | 0.33 ± 0.002 | 0.28 ± 0.004 | 0.36 ± 0.005 |
| Number of independent pulses | 6.30 ± 0.05 | 5.92 ± 0.04 | 5.87 ± 0.03 | 5.82 ± 0.063 | 5.40 ± 0.04 |

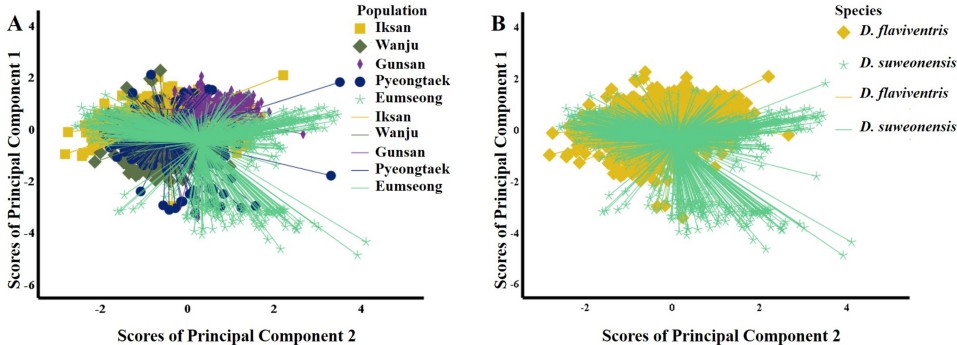

**Figure 3  Scatterplots of the Principal Component Analysis of the call parameters of *Dryophytes suweonensis* sensu lato.** (A) PC1 *vs.* PC2 for five populations. (B) PC1 *vs.* PC2 for two species, *D. flaviventris* and *D. suweonensis*.

## Comparison of the call properties of different populations

To compare the call properties of *D. suweonensis* s.l. from five localities, first, we conducted PCA for the call notes. The PCA identifying the independent dimensions of the call properties between the five populations (all notes) resulted in two PCs, with Eigenvalues of 2.20 and 1.20, respectively, explaining a cumulative variance of 68.03% (Table 1). A variable was judged to be important if displaying a loading factor > 0.55 in one of the PCs. Thus, we excluded high frequency, 90% bandwidth, and number of independent pulses from further analysis (Table 1). The PCs revealed a mixed distribution pattern of variables among the five populations (Fig. 3A). The ANOVA test resulted in insignificant differences within groups and significant differences between groups (Table 1). The *post hoc* Tukey test revealed Gunsan (S) and Eumseong (N) populations were significantly different from others in terms of temporal variables (PC1). In terms of call frequencies (PC2), Iksan (S) and Gunsan (S) had significant differences from all other populations, whereas, Wanju (S) population had no significant difference from Pyeongtaek (N) and Eumseong (N) populations (Appendix 1).

## DISCUSSION

We studied the call properties of *D. suweonensis* s.l. from five localities covering their distribution range to observe the call variations among populations and to see whether it defines species delimitation. Our results suggested variations at the population-level. Many geographically close populations, such as the southern populations Iksan (S), Wanju (S), and Gunsan (S), and the northern populations Pyeongtaek (N) and Eumseong (N) showed significant differences in many call properties. In contrast, many geographically distant populations, such as Pyeongtaek (N) and Iksan (S), Pyeongtaek (N) and Wanju (S), and Eumseong (N) and Wanju (S), showed no significant differences in call properties (Appendix 1). In addition, the call properties did not follow a pattern based on the presumed natural barrier, CM, which *Borzée et al. (2020)* mentioned as a potential geographic barrier for *D. suweonensis* and *D. flaviventris* (Fig. 3B).

In accordance with the previous literature (*Borzée et al., 2020*), our results also indicated the most variability of the call properties of *D. suweonensis* s.l. in the temporal properties (higher Eigenvalues of PC1; Table 1). In the present study, we used delta time, duration of connected pulses, internote interval, and number of independent pulses as the temporal variables of the frog calls. Although *Borzée et al. (2020)* found the number of independent pulses highly important and described it as one of the bases for segregating *D. flaviventris* from the *D. suweonensis*, our study did not detect any significant difference among the populations in this parameter. Despite the variations observed in the number of independent pulses at both individual and population levels, we did not identify a discernable pattern that could be used for species differentiation. Additionally, the variations in the number of independent pulses within the population further supported the concept of individual and population-dependent variations (Fig. 4). During our study, we recorded two to eight Independent Pulses from both southern (*D. flaviventris*; Fig. 4A) and northern populations (*D. suweonensis*; Fig. 4B). Although we observed a slightly higher average number of independent pulses in southern populations ($5.98 \pm 0.021$) than in northern populations ($5.52 \pm 0.03$) like *Borzée et al. (2020)*, the difference was insignificant and inadequate to use it in separating two species.

The average duration of connected pulses for the southern populations in the present study was consistent with that reported by *Borzée et al. (2020*; $0.09 \pm 0.0003$). However, the value was slightly different for the northern populations ($0.09 \pm 0.0006$, which was $0.08 \pm 0.01$ in *Borzée et al., 2020*). Nevertheless, a mixed pattern was observed when we arranged the data according to the population. In terms of the locality, the lowest average duration of connected pulses was observed in the Iksan population (S; $0.08 \pm 0.0007$) followed by the Pyeongtaek population (N; $0.09 \pm 0.0008$) and the highest average duration of connected pulses was observed in the Gunsan population (S; $0.10 \pm 0.0003$) followed by the Eumseong population (N; $0.10 \pm 0.0008$). Likewise, the duration of connected pulses of the calls from the southern and northern populations also exhibited a mixed pattern (Table 2). Moreover, our study did not identify any difference in delta time between populations (Table 2).

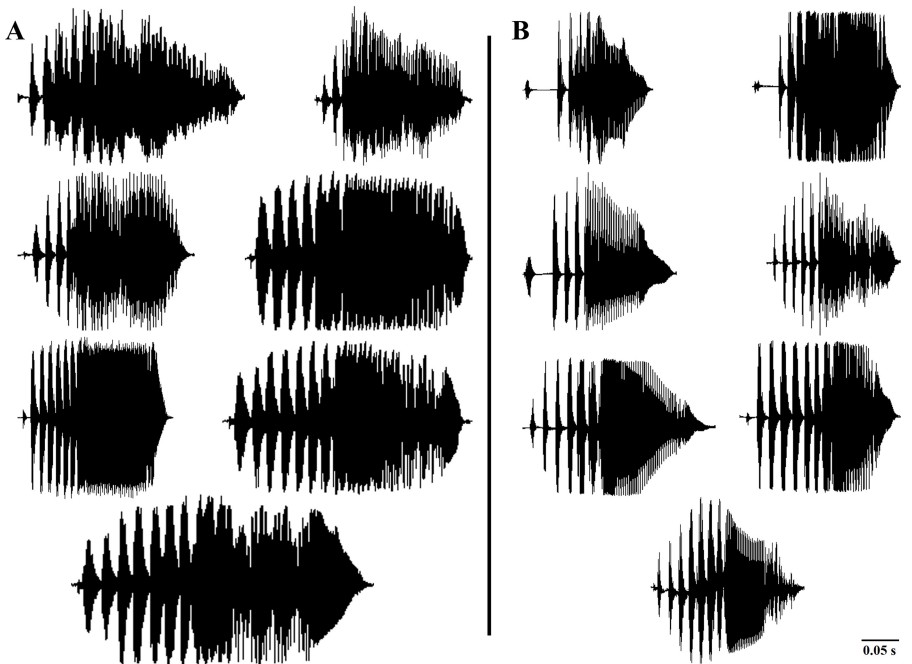

**Figure 4 Variations in the number of independent pulses in the calls of different *Dryophytes suweonensis* sensu lato populations.** (A) Oscillograms of the advertisement calls of the southern populations (*D. flaviventris*). (B) Oscillograms of the advertisement calls of the northern populations (*D. suweonensis*).

The disparities between the results of our analyses and those reported by previous studies might be attributed to the higher number of collected samples and the inclusion of more localities (*Park, Jeong & Jang, 2013*; *Borzée et al., 2020*). Moreover, previous studies categorized the target amphibians as distinct species without conducting population-level analyses (*Park, Jeong & Jang, 2013*; *Borzée et al., 2020*). However, the irregular pattern of variations in call properties, especially the temporal properties, at the population-level is common in many amphibians (*Park & Yang, 1997*; *Forti, Strüssmann & Mott, 2010*; *Forti, Marquez & Bertoluci, 2015*; *Forti, Lingnau & Bertoluci, 2017*; *Briggs, 2010*; *Kaefer & Lima, 2012*; *Kaefer, Tsuji-Nishikido & Lima, 2012*; *Velasquez, 2014*). Some studies have even identified intraspecific complex call patterns (*Porter, 1965*; *Turin, Nali & Prado, 2018*). Furthermore, some researchers have highlighted the influence of local environmental and social conditions on temporal call properties in many amphibian species, discouraging their use in species separation (*Park, Cheong & Yang, 2000*; *Wong et al., 2004*; *Lingnau & Bastos, 2007*).

Although we could not verify this from previous studies (they analyzed data based on species, not locality/population; (*Park, Jeong & Jang, 2013*; *Borzée et al., 2020*), our results did reveal some atypical variations in the high frequency of call properties. The high frequency of the Iksan (S) and Wanju (S) populations was approximately double that of the Gunsan (S) population. Similarly, the high frequency of the Eumseong (N) population was approximately double that of the Pyeongtaek (N) population. However, *Park, Jeong & Jang (2013)* did not find any significant influence of temperature and humidity on the call

properties of *D. japonica* and *D. suweonensis* s.l., which is also supported by our personal data (unpublished). Notably, the Iksan (S), Wanju (S), and Eumseong (N) populations were close to the highway, where there was a constant loud sound of vehicles driving by, and the environmental conditions in all localities were almost similar. Thus, we believe this abnormality might be attributed to noise pollution, a phenomenon that has also been reported in other animals (*e.g.*, *Nemeth et al., 2013*).

## CONCLUSION

Our study revealed an irregular pattern of call properties among populations of *D. suweonensis* s.l., a phenomenon commonly observed in many other amphibians. In addition, the call properties did not follow any specific pattern when considering the populations divided by CM. Considering the goal of this study was only to observe the call properties, we cautiously conclude that the call properties of *D. suweonensis* s.l. likely represent variations at the populations and do not reflect the species-level variations. Thus, we urge further investigation into the specific status of *D. flaviventris* using robust integrated taxonomic approaches including genetic and morphological parameters from a broader array of localities.

## ACKNOWLEDGEMENTS

We extend our gratitude to all lab members who helped in the field work.

### Funding

This work was supported by the Korea Environment Industry & Technology Institute (KEITI) through the Exotic Invasive Species Management Program Program, funded by the Korea Ministry of Environment (MOE; 2021002280004). The funders had no role in study design, data collection and analysis, decision to publish, or preparation of the manuscript.

### Grant Disclosures

The following grant information was disclosed by the authors:
Korea Environment Industry & Technology Institute (KEITI) through the Exotic Invasive Species Management Program Program, funded by the Korea Ministry of Environment (MOE): 2021002280004.

### Competing Interests

The authors declare there are no competing interests.

### Author Contributions

- Md Mizanur Rahman conceived and designed the experiments, analyzed the data, prepared figures and/or tables, authored or reviewed drafts of the article, and approved the final draft.

- Jiyoung Yun performed the experiments, authored or reviewed drafts of the article, and approved the final draft.
- KaHyun Lee performed the experiments, authored or reviewed drafts of the article, and approved the final draft.
- Seung-Ha Lee performed the experiments, authored or reviewed drafts of the article, and approved the final draft.
- Seung-Min Park performed the experiments, authored or reviewed drafts of the article, and approved the final draft.
- Choong-Ho Ham performed the experiments, authored or reviewed drafts of the article, and approved the final draft.
- Ha-Cheol Sung conceived and designed the experiments, authored or reviewed drafts of the article, and approved the final draft.

### Field Study Permissions

The following information was supplied relating to field study approvals (i.e., approving body and any reference numbers):

We did not require permits as we only recorded the calls and did not catch the individuals.

### Data Availability

The call parameters of *D. suweonensis sensu* lato populations from five localities are available in the Supplemental File.

### Supplemental Information

Supplemental information for this article can be found online at http://dx.doi.org/10.7717/peerj.16492#supplemental-information.

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
