# Peer review of "Population-level call properties of endangered Dryophytes suweonensissensu lato (Anura: Amphibia) in South Korea"

_PeerJ, doi:10.7717/peerj.16492_

## Round 0.1 · original submission · Major Revisions

Please revise your paper to address the concerns of the reviewers.

**Language Note:** PeerJ staff have identified that the English language needs to be improved. When you prepare your next revision, please either (i) have a colleague who is proficient in English and familiar with the subject matter review your manuscript, or (ii) contact a professional editing service to review your manuscript. PeerJ can provide language editing services - you can contact us at copyediting@peerj.com for pricing (be sure to provide your manuscript number and title). – PeerJ Staff

Reviewer 1 ·

Basic reporting

This manuscript is about the use of call properties in tree frog species classification. When I checked the original paper, which described the new D. flaviventris, I found that they used very few individuals for each population, such as 3-6 individuals for genetic ddRAD sequences. Other morphological and phonetic characteristics were also obtained from not many individuals, and the analyzed results were largely overlapped among D. immaculatus, D. suweonensis, and D. flaviventris, although some are statistically significant.

Considering the limited sample size for those analyses and statistically significant, but not clear discrimination aspects, D. flaviventris has been in the middle of debates since its publication. The results of this study could give some insights into why one should go over D. flaviventris again based on multiple approaches with much bigger sample sizes. Although this manuscript has several parts that need to be improved and clarified further, the results could be very impactful.

- The first paragraph in the introduction is not largely relevant to this study. They should present some aspects of the recent description and trends of classification of amphibian species using traditional and phylogenetic results.

Experimental design

- The motivation of this study is not clearly presented in the purpose of the study. They should clearly say that they selected those five populations based on Borzee et al. 2020 paper so that they directly compare call properties of two species, previously reported, regardless of the feasibility of previous species classification.

- As far as I know, call properties vary depending on individual conditions and environmental factors such as recording season, recording time in a day, temperature, humidity, etc. I am not assured that the authors have considered those aspects in their call analyses, or at least given an explanation about them. Did they just follow the Borzee et al. 2020 paper, which also did not consider those factors incorrectly?

Validity of the findings

I am not assured that they strongly believe that their results are very conclusive, at least in the aspect of call properties. Are they say that call properties can not be used as a classification key between D. suweonensis and D. flaviventris? It is a crucial part of this paper.

Additional comments

L59, the name is incorrect throughout the text.
L89, explain why the did not analyze mt DNA of their samples.
L92, unfortunately their sample size is not also great.
L102, Capital letter “Connected”
L114, when I ran the PCA analysis using provided authors’ data, I got three PCs, of which eigenvalue is greater than 1, including High Frequency data. Please check the analysis again.
L123, 6.83 should be 7.79.
L126, I don’t understand why the HF in Gunsan is so low compared to other areas. Are any environmental factors during call recording involved in the value? The values in Iksan and Wanju is nearly double to the Gunsan. They should clarify this.

Reviewer 2 ·

Basic reporting

Advertisement call is important to classify the taxa to species level, particularly between cryptic species and sub-species. The author clearly tried to make differences between populations but it does not follow any particular pattern. The introduction should be more tightly connected between the paragraphs. The authors probably do not catch the specimens!! It will be appreciated if the author can add any molecular tools to make differences between the said populations. PCA analyses between populations are also another tool to give some clear idea among the populations.

I have some comments as below:
If possible, please use another species rather than frog which has differences due to the presence of Chilgap Mountain in line 74.
How many calls for each individual was used should be explained in the Descriptive section.

Considering the above suggestion, the paper can be accepted.

Experimental design

The experimental design seems ok.

Validity of the findings

The findings seems ok.

Additional comments

The manuscript can be extended a bit based on some additional findings as per my suggestions.

---

## Round 0.2 · accepted · Accept

Thank you for your careful and conscientious revisions. They adequately addressed the concerns of the reviewers. Congratulations!